# Dominance of *S. cerevisiae* Commercial Starter Strains during Greco di Tufo and Aglianico Wine Fermentations and Evaluation of Oenological Performances of Some Indigenous/Residential Strains

**DOI:** 10.3390/foods9111549

**Published:** 2020-10-26

**Authors:** Maria Aponte, Raffaele Romano, Clizia Villano, Giuseppe Blaiotta

**Affiliations:** 1Division of Microbiology, Department of Agricultural Sciences, University of Naples Federico II, Via Università 100, 80055 Portici, Naples, Italy; aponte@unina.it; 2Division of Food Science and Technology, Department of Agricultural Sciences, University of Naples Federico II, Via Università 133, 80055 Portici, Naples, Italy; raffaele.romano@unina.it; 3Division of Vine and Wine Sciences, Department of Agricultural Sciences, University of Naples Federico II, Viale Italia, 83100 Avellino, Italy; clizia.villano@unina.it

**Keywords:** *Saccharomyces* spp., wild strains, biodiversity, starter cultures, dominance/implantation

## Abstract

In order to evaluate dominance/implantation of starter cultures for wine fermentation, both commercial starters and wild strains were monitored during the fermentation of Greco di Tufo (GR) and Aglianico of Taurasi (AGL) musts. Preliminary characterization of commercial strains was carried out by several molecular markers. Five fermentations—four starter-inoculated and one spontaneous—were carried out in duplicates by using grapes from GR and AGL. Trials were monitored, and yeast cultures were isolated within the dominant microflora. Comparison of Interdelta patterns allowed to assess the real occurrence of both starters and indigenous strains. A high genetic diversity within *S. cerevisiae* strains was detected. In starter-led fermentations (except for few cases), in addition to the starter strains, indigenous *S. cerevisiae* biotypes were found, as well. Native strains isolated from replicates of the same fermentation showed different genetic profiles. *S*pontaneous fermentations were conducted, during the first 5 days, by non-*Saccharomyces* yeasts and, afterwards, by a high number (16 in the AGL and 20 in the GR) of *S. cerevisiae* biotypes. Indigenous biotypes isolated by GR revealed a high variability in oenological features and, in several cases, showed better performances than those recorded for commercial strains. The study further highlighted the low dominance of some commercial starter cultures. Moreover, autochthonous yeast strains proved to be sometimes more aggressive in terms of fermentation vigor in GR must, likely because better adapted to ecological and technological conditions occurring during winemaking. Finally, the use of such strains for production of autochthonous “pied de cuve” may be a useful strategy for lowering production cost of winemaking.

## 1. Introduction

Wine fermentation may be conducted by following two methods. In the starter-led fermentation, commercially-produced yeasts, single- or multi-strain, are used in co-inoculum or in sequential inoculum; in the traditional spontaneous fermentation, yeasts living on grapes and winery surfaces (wild yeasts) carry out the fermentative process [1]. The latter approach is characterized by the succession of several yeast species and strains, with ethanol-tolerant genus *Saccharomyces* spp. dominating the final stages of fermentation [2]. Due to the contribution of a high range of metabolic by-products produced by the complex microbial consortium, spontaneously-fermented wines may be more all-round sensory than those produced by using solely commercial starter cultures [1]. Moreover, it is also widely believed that spontaneous fermentation may impart to wines a more typical character, due to development of a region-specific autochthonous yeast population [3,4]. However, spontaneous fermentations are unpredictable; as matter of fact, slow or arrested fermentations often occur, with subsequent proliferation of spoilage microorganisms [1,5]. On the other hand, the use of commercial starters could mask the distinctive properties of local wines [5]. However, if the starter culture does not fit the specific conditions occurring in the must or the inoculation is not well managed, selected yeasts may not gain dominance, whilst wild yeasts could lead the process due to a better adaptation to the wine cellar environment [1,6]. As matter of fact, the application of strains typing techniques, DNA-based, including Pulsed Field Gel Electrophoresis (PFGE) and restriction analysis of the mitochondrial DNA (mtDNA-RFLP, Restriction Fragment Length Polymorphism), allowed to prove that, in many cases, wild yeasts dominate over inoculated strains [7,8,9]. Therefore, in order to minimize the impact of unwanted ecological evolutions, the wine industry needs to preventively evaluate the inoculated strains’ dominance over the wild microflora. Prevalence needs to be assessed case by case, namely by taking into account variables, such as: starter type, must, and vinification process.

In addition to PFGE and mtDNA-RFLP, microsatellite analysis, minisatellites analysis and Interdelta PCR, and SNP (Single Nucleotide Polymorphisms) analysis, as well as RAD-seq (Restriction Site—Associated Sequencing), have been all used to discriminate between *Saccharomyces* (*S.*) *cerevisiae* strains, as well as to elucidate the strains’ population structure in different winemaking regions [10,11,12,13,14]. Within the above-cited methods, minisatellites, microsatellites (also known as Short Tandem Repeats, STR, or Simple Sequence Repeats, SSR) and Interdelta analyses, are lower-cost methods easily tailored to high-throughput analysis and capable of high discrimination between *S. cerevisiae* strains. Such methods are actually quite popular, having been used in many recent wine and vineyard-associated *S. cerevisiae* studies [13,14,15,16]. Italy stands out worldwide as a wine producer, with great variability in terms of (sometimes very old) cultivars used to produce high-quality wines [17]. Among them, Aglianico di Taurasi and Greco di Tufo are characterized by a high content of proanthocyanidins in the former and by the hint of sulfur often accompanied by other “notes of honey, peach, apricot and dried fruit” in the latter [18]. In particular, ‘Aglianico’ differs by international cultivars, such as ‘Merlot’ and ‘Cabernet Sauvignon’, for its high polyphenol content, high flavanol level, mainly galloylated tannins in seeds and skins and for the high reactivity of grape fractions towards salivary proteins [19,20].

The main aim of the present study was the monitoring of different *S. cerevisiae* commercial starter cultures during the fermentations of musts of “Greco di Tufo” and “Taurasi” DOCG (Appellation of Controlled and Guaranteed Origin), both produced in the Irpinia district of Campania Region (Italy) in order to evaluate their dominance over natural yeast microflora. Moreover, basic technological performances of natural occurring *S. cerevisiae* strains (from “Greco di Tufo”) were compared with those of commercial strains with the aim to select good candidates to use as starter culture for the production of Greco di Tufo DOCG wine.

## 2. Materials and Methods 

### 2.1. Identification and Characterization of Commercial Starter Strains 

Eight *S. cerevisiae* commercial active dry starters currently used for winemaking in the Irpinia district were purchased from a specialized local shop. Four strains, herein named W1, W2, W3, and W4, were used for Greco di Tufo (White) fermentations, and four strains, herein named R1, R2, R3, and R4, were used for Aglianico of Taurasi (Red) fermentations. Strain coded with “W” are sold for white wine fermentations and those coded with “R” for red wine fermentations. A small amount of each active dry culture was rehydrated and activated in 10 mL of YPD (Yeast extract, Peptone, Dextrose) medium (Yeast extract 10 g, Bacteriological Peptone 20 g, glucose 20 g, pH 6.5) All components were provided by Oxoid (Basingstoke, UK). Strains were checked for purity by repetitive streaking onto WL Nutrient Agar plates (Oxoid) and stored at −80 °C in the same medium added of glycerol (25% *v*/*v*).

DNA from starter strains was isolated by following the protocol previously described by Aponte and Blaiotta [14]. Briefly, cells from YPD broth cultures in exponential growth were washed and resuspended in 50 mM Na_2_EDTA (pH 7.5). Five mL of 50 mM Tris-H_2_SO_4_, pH 9.3 and 0.5 mL of 1% 2-mercaptoethanol were added, as well. After incubation for 15 min at room temperature, followed by centrifugation at 180× *g* for 5 min, 500 µL of 0.1 M NaCl/10 mM tris-HCl, pH 7.5/10 mM Na_2_EDTA/0.2% SDS were added to the pellet. After incubation at 65 °C for 15 min, 1.2 vol of potassium acetate 3M were added. After cooling on ice for 30 min, and centrifugation at 4 °C (1400 rpm for 10 min), the supernatant was added of an equal volume of iced isopropanol. The mixture was allowed to rest for 5 min at room temperature, and, after centrifugation, the pellet was washed with 300 µL of iced ethanol (70%). The DNA pellet was resuspended in 50 µL of 10 mM Tris-H_2_SO_4_, 1 mM Na_2_EDTA (pH 8.8) at 65 °C for 30 min, and then treated with 2 µL of 10 mg/mL DNase free-RNase at 37 °C for one hour. Two *S. cerevisiae* strains (NA157 and SB-C) from the collection of the Division of Vine and Wine Sciences were included as controls.

Taxon of commercial strains was confirmed by ITS-RFLP (Internal Transcribed Spacer–RFLP) as previously described by Aponte and Blaiotta [14]. All yeast cultures were subject to strain typing by means of Interdelta analysis, *DAN4* minisatellites analysis and mtDNA-RFLP with *Rsa*I endonucleasis as previously described by Aponte and Blaiotta [14].

### 2.2. Wine Fermentation Protocols

Grapes (*Vitis vinifera* cultivars Greco and Aglianico) used in this study were provided by farms located in the DOCG production areas of “Greco di Tufo” (GR) and “Taurasi” (AGL) (Irpinia district). 

The vinification protocols described by De Filippis et al. [21] were applied. Potassium metabisulfite (10 g/100 kg) was added to GR grapes after crushing and de-stemming, whilst 5 g/100 kg of Vitamin C were added before pressing. Drained must was clarified at 4–6 °C for 24 h and fermented in duplicate (80 L each) in a 100 L thermo-controlled steel container at 18–20 °C. After crushing and de-stemming, AGL grapes were added of 10 g/100 kg of potassium metabisulfite and fermentation was carried out in duplicate (80 L each) at 23–25 °C. Fermentations were all performed at the Division of Grape and Wine Sciences, but at different times. Nutristart (Laffort s.r.l., Tortona, Italy), containing di-ammonium phosphate, inactive dry yeast, yeast autolysate and thiamine, was used as fermentation activator (100 mg/L provides 14 mg/L of total available nitrogen and 0.13 mg/L of thiamine). In GR, Nutristart was added at the beginning of fermentations (10 g/hL), as well as after 4 and 9 days (5 g/hL). In AGL, it was added at the beginning and after 4 days (10 g/hL each). Active dry cultures were aseptically rehydrated as recommended by suppliers and used at the dose of 20 g/hL. All fermentations—four starter-inoculated plus controls—were performed in duplicate and constantly subject to physico-chemical and microbiological analyses. Samples (200 mL) were collected at time 0, immediately after starter adding and after 5, 10, and 20 days for GR. In AGL fermentations, the first samples were collected at about 12 h after the starter adding, as well as during the first mixing (0.5 days) and then at 6.5 and 12.5 days.

### 2.3. Microbial Counts, Yeasts Isolation, and Strain Typing

Musts and wines were serially diluted in quarter strength Ringer’s solution (Oxoid) and spread-plated on WL-nutrient agar (Oxoid). After incubation at 28 °C for 5 days, plates were used for viable counts and yeasts isolation. WL is a non-selective medium that allows to discriminate wine yeasts species and strains and can be profitably used for monitoring the yeast population dynamics during wine fermentation [14]. In order to analyze the dominant cultivable yeast microbiota, colonies showing different morphology and color on plates with 15–150 colonies were isolated and purified on WL nutrient agar. In the isolation phase, an attempt was made to maintain the proportions of the different types of colonies.

DNA was extracted according to Aponte and Blaiotta [14]. All isolates were subject to *Hae*III ITS-RFLP analysis. Strains referable to *S. cerevisiae* were typed by Interdelta analysis and by *DAN4* Minisatellite analysis [14]. DNAs isolated from starter cultures were constantly used as control. Fingerprinting images were compared by means of Bionumerics software (v. 5.1, Applied Maths, Kortrijk, Belgium) and similarities among patterns were assessed. Profiles showing more than 95% of similarity were considered identical.

### 2.4. Technological Characterization of Strains Isolated from Greco Fermentations

The four starter strains (W1, W2, W3, W4), plus forty-three Interdelta-different *S. cerevisiae* strains, were recovered from ten fermentations of Greco di Tufo must and further evaluated for biochemical features of technological interest. Ethanol tolerance was assessed in YPD broth containing ethanol at six increasing concentrations (10, 11, 12, 13, 14, and 15% *v*/*v*). Cultures were pre-adapted by cultivation in the same medium added of 7% ethanol. After incubation at 20 °C for 72 h, growth was assessed by spectrophotometry at white light (600 nm). Sulfur dioxide (SO_2_) tolerance was evaluated in YPD broth adjusted at pH 3.5 with tartaric and malic acids (1:1) and supplemented with two potassium metabisulfite concentrations (100 and 200 mg/L). Hydrogen sulfide (H_2_S) production was estimated on Biggy agar (Oxoid) after incubation at 28 °C for 5 days. For browning description, the following codes were used: low production, snow-white color (<2); medium production, hazelnut-brown color (2–3); and high production, rust-coffee color (3–5) [14]. 

Antagonistic activity was assessed according to Sangorrin et al. [22], by using *S. cerevisiae* CECT1890 as sensitive strain. β-glucosidase activity was evaluated in a medium containing cellobiose (7.5 g/L), peptone (2.5 g/L), and bromothymol blue (0.06 g/L): growth and pH lowering were associated to enzymatic activity. Esterase activity was evaluated on Tween 80 as described by Slifkin [23]: the formation of a calcium precipitate around colonies provided indication of esterase activity. Growth performances at low temperature (14 °C) and type of growth were evaluated by using synthetic must (230 g/L of total sugars, tartaric acid 4 g/L, malic acid 2 g/L, pH 3.2) according to a protocol previously detailed by Rinaldi et al. [24]. All tests were done in duplicate.

In order to estimate the percentage of similarity among isolates, data were subject to cluster analysis (Average Linkage Method). A correlation matrix was constructed using the formula described by Upholt [25] and Nei and Li [26]: Fxy = (2n_xy_)/(n_x_ + n_y_), where F_xy_ is the proportion of common technological traits of compared isolates (x and y), n_xy_ is the number of characters shared by both isolates x and y, and nx and ny are the total of number characters observed in isolates x and y, respectively. The resulting correlation matrix was analyzed by Systat 5.2.1 software.

### 2.5. Fermentation Performances of Selected Yeast Strains

Fermentation vigor (FV) and fermentation power (FP) were evaluated by means of micro-fermentation trials in Greco di Tufo must (°Brix 20.5, pH 3.32; total acidity 10.08 g/L of tartaric acid) added of metabisulfite (100 mg/L). Strains, cultured twice in YPD medium, were used to inoculate at about 6 Log CFU/mL, 80 mL of tyndallized (100 °C for 3 min for 3 times) must in 250 mL Erlenmeyer flasks closed with a Müller valve filled with sulfuric acid. During incubation (at 18 °C), flasks were handle-stirred for 30 s every 12 h. Weight loss, due to CO_2_ escaping from the system, was quantified by monitoring the fermentation kinetics. Fermentation was considered concluded when no weight loss was any longer recorded within 24 h. FV was expressed as grams of CO_2_ produced in 80 mL of must during the first 72 h of fermentation, while FP was expressed as grams of CO_2_ produced until the end of fermentation. Each trial was performed in triplicate. 

Color intensity and tonality were evaluated by reading the absorbance at 420, 520, and 620 nm (Eppendorf Basic BioSpectrometer, Italy) of wine samples after centrifugation at 14,000 rpm for 5 min. The color intensity was calculated as the sum of absorbance A420, A520, and A620 values, while the color tonality was assessed by the ratio A420/A520.

### 2.6. Chemical Determinations

The pH was determined by using a lab pH-meter (XS, model pH 50, Modena, Italy). Total acidity of samples was estimated by titration method (25 mL of wine sample and 0.25 N NaOH) and expressed in g/L of tartaric acid. Concentrations of acetic acid, total sugars (glucose plus fructose), glycerol, and ethanol were evaluated by HPLC (High Performance Liquid Chromatography) as previously described [14].

### 2.7. Statistical Analysis

Significant differences among microbiological and oenochemical parameters of AGL and GR musts/wines at different fermentation times, as well as of wines obtained by micro-fermentation experiments, were computed by using ANOVA and Tukey *t*-test (*p* < 0.05) (XLStat 2012.6.02 statistical pocket, Addinsoft Corp., Paris, France).

## 3. Results

### 3.1. Commercial Yeast Strains Identification and Genetic Characterization

According to ITS analysis, all strains proved to belong to the genus *Saccharomyces* sensu stricto since the size of the amplicon was of 850 bp. After *Hae*II ITS-RFLP analysis, they were confirmed as belonging to *S. cerevisiae* species. All strains were then subject to a molecular characterization by means of three different techniques: Interdelta analysis, minisatellite *DAN4*, and *Rsa*I mtDNA-RFLP. All approaches were carried on twice to check the reproducibility of the method even using different DNA concentrations. Results of Interdelta analyses are reported in Figure 1. The method proved to be very reliable, since patterns of the same strains were always perfectly overlapping. Moreover, all analyzed strains appeared completely discriminated since profiles proved to be strain-specific (Figure 1). Indeed, *DAN4* minisatellite analysis and mtDNA-RFLP-*Rsa*I were discriminating, as well: patterns were characterized by a low number of bands closely positioned (Appendix A). On the other hand, the mtDNA-RFLP-*Rsa*I produced somewhat different profiles but was rather difficult to read (Appendix A).

### 3.2. Starter Tracking during “Greco di Tufo” and “Aglianico Taurasi” Microvinifications

Yeast loads were monitored by counting on WL agar. Results for the two independent replicates are reported in Table 1 and Table 2. As expected, at the beginning of monitoring, the yeast population’s level was always higher than 10^6^ CFU/mL, except for controls which were, in both cases, almost one Log lower (Table 1 and Table 2). 

In GR fermentations, yeast reached the same level (about 7.5 log CFU/mL) after 5 days in all batches and then stayed almost stable up to 10 days. By this point on, loads decreased, but some differences emerged within the five trials; in fact, in vinifications accomplished by the inoculation of strain W3, yeast levels remained significantly higher (Table 1). A low variability within trials could be recorded by turning to AGL microvinifications (Table 2). Except for one of the four starter-inoculated fermentations (R2), the population level, at 12 h from starter addition, was higher than 10^7^ CFU/mL. In spontaneous fermentations, at the same time, yeast counts were significantly lower (5.36 ± 0.69 Log CFU/mL; Table 2). After 6.5 days of fermentation, yeasts reached about 10^7^ and onward little decreased at 6.28–7.02 Log CFU/mL. No statistically significant (*p* < 0.05) differences were recorded between starter-inoculated fermentations and controls at both 6.5 and 12.5 days (Table 2).

### 3.3. Yeast Strains Identification

One hundred and eighty-two and 119 yeast cultures were isolated from the WL agar counting plates used for the vinifications monitoring of GR and AGL, respectively. Cultures were chosen by taking into account colonies appearance and pigmentation. By *Hae*III ITS-RFLP analysis, 163 out of 182 strains (89.5%) collected during the fermentation of GR could be reported to the *S. cerevisiae* species (Table 3). At beginning of the process, non-*Saccharomyces* proved to be part of the dominant microflora, even in six out of the eight starter-inoculated trials (Table 3). As expected, in both control fermentations, all collected strains at time zero were non-*Saccharomyces* (Table 3). ITS sequencing showed that several non-*Saccharomyces* species occurred: *Metschnikowia* (*M.*) *pulcherrima*, *Hanseniaspora* (*H.*) *uvarum*, *Pichia* (*P.*) *occidentalis*, *P. anomala*, and *P. fermentans.*

At any rate, all colonies collected by the day 5th up to the end of monitoring—20 day in trials with GR must—could be reported to the genus *Saccharomyces* in sensu stricto.

In the AGL fermentations, a total of 119 yeast cultures were isolated. Ninety-three cultures—78%—were reported to the *S. cerevisiae* species by *Hae*III ITS-RFLP. As reported in Table 3, 24 non-*Saccharomyces* were retrieved at 12 h from the inoculum, and even after almost one week, in just two cases: AGL7 (W1) and AGL5 (control). In all inoculated batches, at the beginning of monitoring, with the exception of AGL3 (R3), non-*Saccharomyces* were dominant or co-dominant (Table 3). Non-*Saccharomyces* retrieved at beginning were represented by *H. uvarum*, *M. pulcherrima*, *P. fermentans*, and *Starmerella* (*St.la*) *bacillaris*. This last species was retrieved as co-dominant after 6.5 days of fermentation in batches AGL7 (R1) and AGL5 (control).

### 3.4. Strains Tracking by Molecular Typing

All the yeast cultures reported to the genus *Saccharomyces* by ITS length were further subject to Interdelta analysis for evaluating the starter survival during the fermentations. Strains showing the same Interdelta pattern of the starter culture were always coded as “a” (Table 4) and, however, confirmed by *DAN4* minisatellite. Obviously, starters were always part of the dominant microflora in all the inoculated vinifications. In GR, strains with profiles different from that of the starter occurred in inoculated fermentations, as well. In GR8 (W2) and GR3 (W3), indigenous *S. cerevisiae* strains were retrieved already at beginning of fermentation, while, in all other cases, wild *Saccharomyces* were recovered by day 5 up to the end of monitoring. Constantly, different indigenous *S. cerevisiae* biotypes were detected at each specific phase of the process. In fact, only in two cases, the same biotype was retrieved in more than one phase: biotype “e” in GR3 (W3) at 10 and 20 days; biotype “f” in GR10 (W4) at 5 and 10 days. In spontaneous fermentations GR5 and GR6, a high number of different *S. cerevisiae* Interdelta-biotypes could be noticed by the day 5 up to day 20 (Figure 2). In every phase of both replicates, Interdelta biotypes different from the one of the starter were always detected (Table 4).

In starter-led fermentations coded as AGL7 and AGL8, the inoculated *S. cerevisiae*—namely R1 and R2, respectively—were not retrieved at the first sampling (0.5 days) (Table 4). By the second sampling point (t6.5) to the end of fermentation (t12.5), the inoculated strains dominate or co-dominate with indigenous *S. cerevisiae* biotypes (Table 4). In fermentations with starter R4 (AGL4 and AGL10), a high incidence of indigenous *S. cerevisiae* biotypes was recorded in each phase. Specifically, biotype “b” found at t0.5 in vat AGL10 (R4) was found at t6.5 and t12.5 in vat AGL4 (R4), too (Table 4). In the two natural AGL fermentations, a total of 14 different biotypes were detected (a-p). In detail, in the control fermentation AGL5, a biotypes’ succession seems to occur: biotypes “a”, “b”, “c” at 6 days and “d”, “e”, “f”, “g” at 12 days (Table 3). In the trial AGL6, two patterns were detected at 6 days (“h” and “i”), as well as at the end of fermentation after 12 days (Table 4).

### 3.5. Technological Characterization of Yeast Cultures from Greco di Tufo

Forty-three *S. cerevisiae* biotypes obtained from the ten GR experimental vinifications plus four commercial starter strains (W1, W2, W3, W4) were characterized under a technological point of view (Table 5). All strains were able to grow in YPD broth at pH 3.50 containing 200 mg/L of potassium metabisulfite. No strains exhibited esterasic activity against Tween 80, or β-glucosidasic hydrolysis of cellobiose. Eighteen strains (including W3 and W4) out of 47 were able to grow in presence of 15% ethanol. Of the remaining strains, 20 (including W1) were resistant at 14%, whilst eight (including W2) were at 13% and one at 12%. On the basis of the H_2_S production, strains were gathered in three groups: 15 strains were low producers (level < 2); 21 strains (including W1, W2 and W3) were medium producers (level 2–3); 11 strains (including W4) were high producers (level 3–5). The killer factor proved to be quite widespread within strains; in fact, 18 wild strains, plus three commercial cultures (W1, W2, and W3), displayed this trait. According to obtained outcomes, only 18 strains (38%), including W1 and W4, were able to low the °Brix of at least three points after five days of fermentation. Finally, just 15 cultures exhibited a dispersed type of growth at the tested conditions (Table 5). On the basis of cluster analysis, at 70% of similarity, strains could be gathered in 28 groups (Figure 3). A total of 21 strains (including W1, W2, W3 and W4) were tested in micro-vinification experiments in sulfated Greco di Tufo must (20.5 °Brix). Fermentation performances of selected wild *S. cerevisiae* strains versus commercial strains are summarized in Table 6.

Even if strains proved to be quite similar in terms of FP (range 7.08–7.84), a high diversity with reference to FV could be recorded (range 0.77–1.60). Fourteen cultures (66%), including three commercial and 11 wild strains, showed FP values higher than 7.6. Surprisingly, commercial strains showed low FV values (0.80–1.05). By contrast, more than 50% of the wild strains displayed FV values higher than 1.2.

Despite non-significant differences of wines’ ethanol level, the residual sugar content was very variable: eight wines (produced by 2 commercial and 6 wild strains) were dry (<2 g/L), and 7 (produced by 1 commercial and 6 wild strains) were very sweet (>5 g/L). In all wines, an acceptable level of acetic acid was detected (<0.5 g/L), even if only in two wines (W1 and GR4-T5-61), it was even very low (0.15 g/L). The glycerol content of wines ranged from 4.4 to 6.9 g/L; however, only 3 strains produced wines with more than 6.0 g/L (W1, GR5-T5-62, GR3-T20-55).

Even if there were non-significant differences in wines’ pH, some differences were recorded with reference to total acidity, which ranged from 9.5 to 12.0 g/L. The lowest total acidity values were detected in wines produced by using W2, as well as in those produced with five wild strains. On the other hand, the highest total acidity value was reported for the wine produced by strain GR4-T20-53, which actually showed the highest sugars content, as well.

Clearly evident differences in terms of color attributes of wines could not be noticed, with the exception of wines obtained with W3 (I 1.40 and H 3.91) and GR5-T10-63 (I 0.87 and H 4.36), which showed the lower and the higher values.

## 4. Discussion

*S. cerevisiae* is the key microorganism of wine fermentation. Due to its absence or low occurrence on grape berries is considered an opportunistic species of the berry microbiota [27]. In fact, experiments performed on sterile collected grapes and fermented in sterile bags showed that, only in nine out of 12 samples, *S. cerevisiae* become dominant and is therefore isolable [28]. The study of Goddard et al. [29] evidenced the local dispersal of *S. cerevisiae* by insects and the global dispersal by human in winery environment. Colonization of winery surfaces by *Saccharomyces* has been reported by different researchers so far, but only the study of Bokulich et al. [30] quantitatively demonstrated that *Saccharomyces* is an abundant and dominant component of pre-harvest population on winery surfaces. Moreover, Martiniuk et al. [13] proved that the winery *S. cerevisiae* may significantly alter the initial yeast composition of the must. Authors also highlighted the high abundance and diversity of commercial-related strains in winery fermentations, suggesting that yeast population descended by commercial strains may become resident on winery facilities [13]. *S. cerevisiae* cellar-residential strains were also retrieved in our previous study [14]. The repetitive use of commercial yeasts may thus decrease the diversity and the incidence of indigenous *S. cerevisiae* populations in winery fermentations [31]. Lange et al. [32] found that, when the starter’s implantation/persistence is relatively low, the composition of *S. cerevisiae* populations resembles a spontaneous fermentation, rather than a fermentation conduced in wineries which extensively used commercial ADY (Active Dry Yeast) as inoculants. The observation that some commercial strains may consistently dominate fermentations, even when they are no longer used as an inoculant, suggests that an intense use of aggressive commercial *S. cerevisiae* strains allows them to become residential in the vinery. Such prevalence reduces both inoculants and indigenous *S. cerevisiae* strains occurrence during fermentation [31,32]. Some studies proved that indigenous, more than commercial *S. cerevisiae* strains, may out-compete with the inoculant [31]. As a consequence, spontaneous fermentations are characterized by a varied succession of yeasts, however, by the middle of process, one or more strains of *S. cerevisiae* (from grapes or cellar) dominate the fermentation [13,14,21,28], as reported in the present study. In starter-led fermentations, mainly due to higher initial loads of inoculated *Saccharomyces* strains over the natural occurring ones, the starter is supposed to dominate the fermentation process. Delteil [33] and Clavijo et al. [34] suggested that the inoculation process has to be considered unsuccessful when the implantation percentage is less than 50–60%. Lange et al. [32] showed variable, and often low, levels of dominance/persistence of the inoculated strains, depending on commercial starter used, as well as of the winery. In this study, different levels of dominance were observed depending on commercial starter used and on the fermentation stage. In Greco, which is a white must, dominance ranged from 45.5 to 100% during fermentation, but, at the end the process, was less than 80% in all cases. In Aglianico, a red wine, three out four strains showed dominance levels higher than 80%, while the fourth strain (R4) showed a progressive reduction of occurrence along fermentation: from 90% to 25%, at beginning and at the end of monitoring, respectively. However, in R4 fermentations, indigenous *Saccharomyces* strains were detected even at the initial stages of the process. In addition, *St.la bacillaris* was retrieved as co-dominant after 6.5 days of fermentation in batches AGL7 (R1) and AGL5 (control). Strains of this species, already isolated in Aglianico grapes, showing a fructophilic nature, high glycerol production, relative low ethanol and acetic acid synthesis, and partial malic acid metabolization [28] may represent a valid instrument to modulate some wine characteristics.

Therefore, considering that the competition degree of each strain is influenced by both abiotic and biotic factors, which determine the capacity of a strain to out-compete [35], the choice of the starter should take into account several aspects, including must characteristics and vinification technology.

By the analysis oenological traits and fermentation performances of natural *S. cerevisiae* strains occurring in Greco fermentations and co-dominant with the inoculated strain, showed a high diversity. Moreover, some wild isolates exhibited promising technological features. Particularly, fermentation performances, evaluated in the same must adopted for the isolation, were sometimes better than those recorded for the starter strains. The fermentation vigor, namely the speed of fermentation initiation, of natural strains was often much higher (even 1.6) than those reported for commercial starter strains (max 1.0). The occurrence of such aggressive strains is certainly an important biotic factor that could adversely affect the implantation of commercial starter strains during fermentation. Results here collected further confirm the hypothesis of Capece et al. [36,37] in that the autochthonous yeast strains are likely better adapted to the ecological and technological conditions of their own winegrowing area. Finally, winemakers are increasingly keen to limit the use of commercial yeasts in order to reduce oenological inputs and to increase the sustainability of the wine sector. Spontaneous fermentation is certainly a more sustainable process than controlled fermentation [38]. However, given its unpredictability in recent years it is not widely adopted. To obviate the unpredictability problems of spontaneous fermentation and in any case make the fermentation process more sustainable two methods were recently purposed by Borlin et al. [39]: (i) the selection of high fermentative *S. cerevisiae* yeasts from the indigenous population, followed by small scale industrial production; (ii) the preparation of indigenous ‘pied de cuve’ (PdC). Even if the PdC use is becoming popular, especially in organic farming systems, its preparation methods are still empirical procedures [39,40] and sometimes do not provide good results. Therefore, the use of indigenous selected strains for PdC production may be a profitable strategy for both reduce costs and obtain more certain results.

## 5. Conclusions

The choice of the strains to be used as a starter for wine fermentation is of fundamental importance. The dominance of commercial strains in specific contexts should be assessed in advance.

The strategy to select starters within the indigenous microflora, and thus better adapted to fermentation conditions of specific musts, could counteract the low dominance/implantation of conventional starter cultures for wine fermentation. In fact, the strain (GR5-T10-63), isolated during natural fermentation of Greco di Tufo must, and exhibiting good ethanol tolerance (14%), low H_2_S production, excellent fermentation vigor, and good fermentation power, may be a good candidate as starter culture for the production of Greco di Tufo DOCG wine.

## Figures and Tables

**Figure 1 foods-09-01549-f001:**
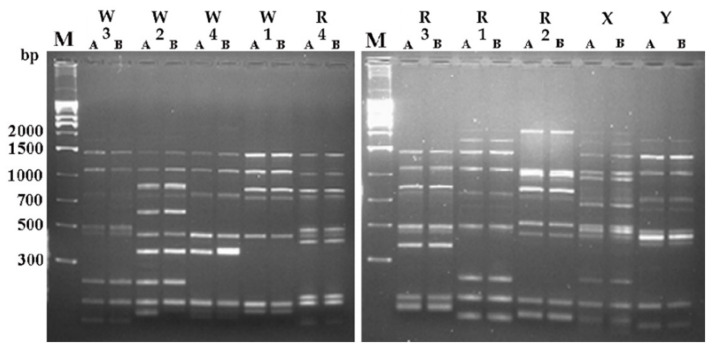
Interdelta patterns of commercial starter strains used in this study. A and B are replicates of the same strain. X and Y are *S. cerevisiae* strains belonging to the personal collection. W and R are strains used for Greco di Tufo (White) and Aglianico of Taurasi (Red) fermentations, respectively. M: marker 1 kb Ladder plus (Invitrogen^TM^)

**Figure 2 foods-09-01549-f002:**
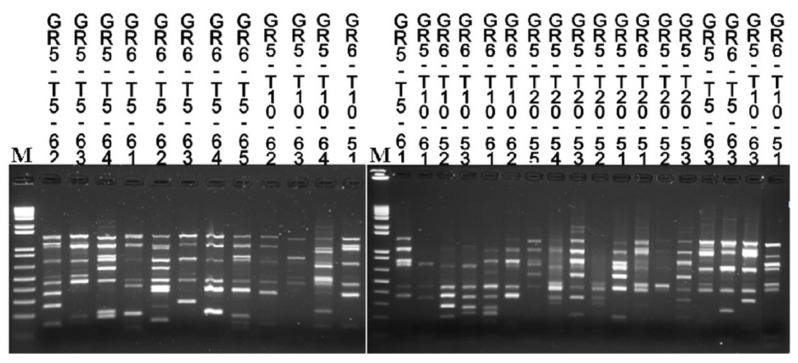
Interdelta patterns of selected *S. cerevisiae* strains isolated from natural fermentations Greco di Tufo (GR)5 and GR6. T5, T10 and T20: strains isolated after 5, 10 and 20 days of fermentation.

**Figure 3 foods-09-01549-f003:**
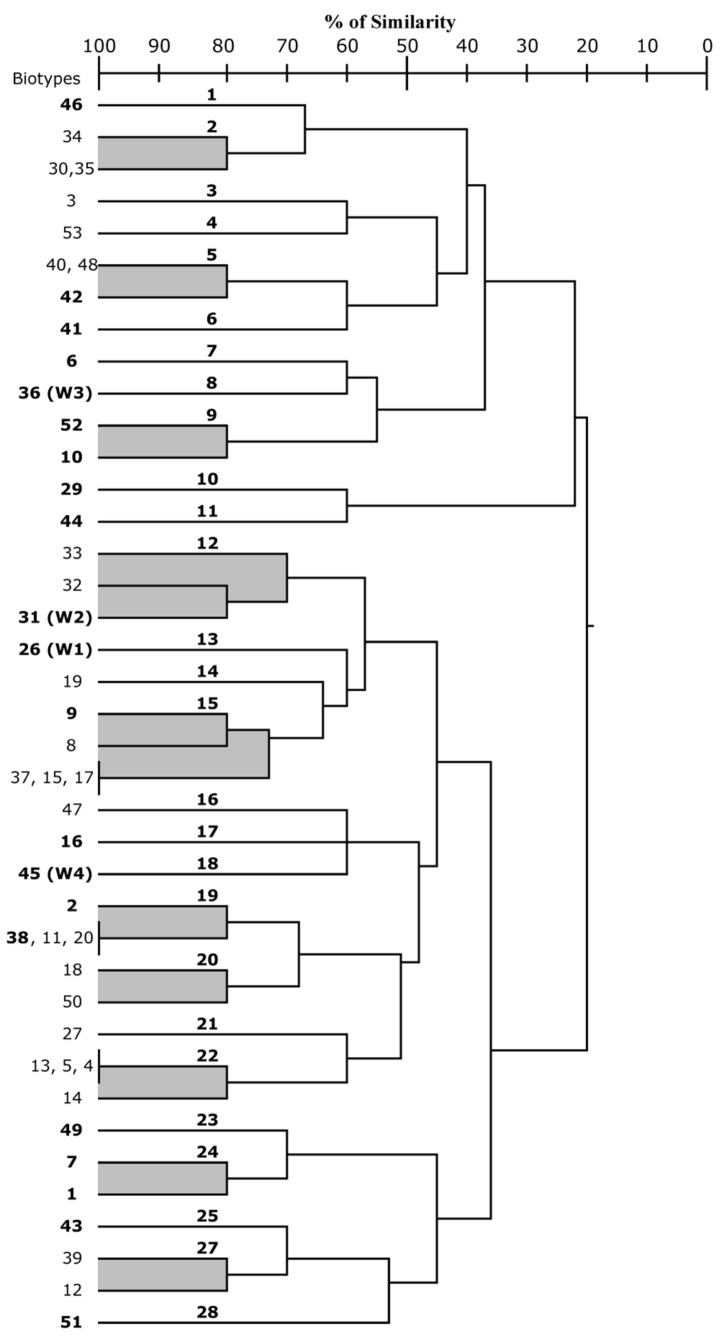
UPGMA Unweighted Pair Group Method with Arithmetic mean) dendrogram obtained from the comparison of some technological traits (see Table 5). Clusters were defined at 75% of similarity. Biotypes in bold case are those selected for micro-fermentation trials (Table 6).

**Table 1 foods-09-01549-t001:** Monitoring microbiological and oenochemical parameters during “Greco di Tufo” (GR) experimental fermentations (data are mean ± standard deviation of two independent fermentations).

Fermentations (Starter Strain)	Time (Days)	Microbiological and Oenochemical Parameters						
		Yeast Loads ^1^	pH	TA ^3^	TS ^3^	ET ^3^	GLY ^3^	AA ^3^
GR1/GR7 (W1)	0	6.78 ± 0.02 ^a 2^	3.29 ± 0.01 ^a^	5.95 ± 0.11 ^a^	218.51 ± 0.16 ^a^	nd ^4^	nd	nd
GR2/GR8 (W2)		6.77 ± 0.01 ^a^	3.30 ± 0.01 ^a^	6.03 ± 0.04 ^a^	218.37 ± 0.06 ^a^	nd	nd	nd
GR3/GR9 (W3)		6.22 ± 0.23 ^b^	3.32 ± 0.02 ^a^	5.92 ± 0.17 ^a^	218.08 ± 0.57 ^a^	nd	nd	nd
GR4/GR10 (W4)		6.38 ± 0.04 ^ab^	3.29 ± 0.01 ^a^	6.05 ± 0.08 ^a^	217.80 ± 0.52 ^a^	nd	nd	nd
GR5/GR6 (Control)		5.24 ± 0.06 ^c^	3.30 ± 0.01 ^a^	6.01 ± 0.01 ^a^	217.60 ± 0.08 ^a^	nd	nd	nd
GR1/GR7 (W1)	5	7.61 ± 0.06 ^a^	3.25 ± 0.00 ^a^	7.18 ± 0.01 ^bc^	141.52 ± 0.59 ^c^	4.20 ± 0.02 ^ab^	3.47 ± 0.21 ^b^	0.13 ± 0.01 ^b^
GR2/GR8 (W2)		7.63 ± 0.03 ^a^	3.25 ± 0.00 ^a^	7.10 ± 0.01 ^cd^	136.58 ± 0.27 ^d^	4.53 ± 0.08 ^a^	3.85 ± 0.07 ^ab^	0.09 ± 0.01 ^c^
GR3/GR9 (W3)		7.66 ± 0.05 ^a^	3.25 ± 0.01 ^a^	7.30 ± 0.01 ^a^	145.78 ± 0.58 ^b^	3.97 ± 0.00 ^b^	3.59 ± 0.08 ^ab^	0.16 ± 0.01 ^a^
GR4/GR10 (W4)		7.50 ± 0.06 ^a^	3.27 ± 0.01 ^a^	7.07 ± 0.05 ^d^	154.26 ± 54.2 ^a^	3.43 ± 0.17 ^c^	3.62 ± 0.03 ^ab^	0.18 ± 0.01 ^a^
GR5/GR6 (Control)		7.61 ± 0.02 ^a^	3.25 ± 0.00 ^a^	7.23 ± 0.00 ^ab^	154.66 ± 0.91 ^a^	3.39 ± 0.11 ^c^	3.97 ± 0.13 ^a^	0.15 ± 0.00 ^ab^
GR1/GR7 (W1)	10	7.56 ± 0.03 ^ab^	3.28 ± 0.00 ^b^	7.51 ± 0.43 ^ab^	27.89 ± 1.90 ^b^	10.50 ± 0.50 ^ab^	6.97 ± 0.12 ^cd^	0.21 ± 0.04 ^ab^
GR2/GR8 (W2)		7.58 ± 0.00 ^ab^	3.32 ± 0.01 ^a^	7.32 ± 0.06 ^b^	19.92 ± 2.95 ^b^	11.17 ± 0.33 ^a^	7.39 ± 0.05 ^bc^	0.14 ± 0.06
GR3/GR9 (W3)		7.64 ± 0.05 ^a^	3.27 ± 0.00 ^b^	8.24 ± 0.01 ^a^	40.85 ± 4.87 ^a^	9.51 ± 0.24 ^b^	6.80 ± 0.01 ^d^	0.19 ± 0.01 ^ab^
GR4/GR10 (W4)		7.50 ± 0.01 ^b^	3.32 ± 0.01 ^a^	7.69 ± 0.01 ^ab^	28.39 ± 0.57 ^b^	10.27 ± 0.07 ^ab^	7.52 ± 0.20	0.29 ± 0.00 ^a^
GR5/GR6 (Control)		7.58 ± 0.04 ^ab^	3.30 ± 0.01 ^ab^	8.14 ± 0.05 ^a^	27.24 ± 0.95 ^b^	10.25 ± 0.01 ^ab^	8.21 ± 0.01 ^a^	0.22 ± 0.00 ^ab^
GR1/GR7 (W1)	21	6.32 ± 0.09 ^b^	3.30 ± 0.00 ^bc^	7.55 ± 0.09 ^b^	4.84 ± 0.12 ^ab^	12.68 ± 0.06 ^abc^	7.39 ± 0.15 ^c^	0.19 ± 0.01 ^c^
GR2/GR8 (W2)		6.07 ± 0.08 ^b^	3.32 ± 0.01 ^ab^	7.50 ± 0.01 ^b^	3.10 ± 0.88 ^b^	12.84 ± 0.07 ^a^	7.59 ± 0.06 ^bc^	0.10 ± 0.00 ^d^
GR3/GR9 (W3)		6.77 ± 0.04 ^a^	3.29 ± 0.01 ^c^	8.01 ± 0.08 ^a^	7.68 ± 1.80 ^a^	12.45 ± 0.01 ^c^	7.44 ± 0.06 ^c^	0.20 ± 0.01 ^bc^
GR4/GR10 (W4)		6.17 ± 0.03 ^b^	3.33 ± 0.00 ^a^	7.33 ± 0.07 ^b^	5.30 ± 0.95 ^ab^	12.58 ± 0.08 ^bc^	8.00 ± 0.15 ^b^	0.31 ± 0.01 ^a^
GR5/GR6 (Control)		6.14 ± 0.09 ^b^	3.32 ± 0.00 ^a^	7.83 ± 0.01 ^a^	3.24 ± 0.62 ^b^	12.77 ± 0.04 ^ab^	8.62 ± 0.04 ^a^	0.22 ± 0.01 ^b^

^1^ Log CFU/mL (determined on WL Nutrient Agar; Oxoid). ^2^ Data significance for columns and sampling time (ANOVA: Tukey *t*-test. *p* < 0.05. XLStat). ^3^ TA: total acidity (g/L of tartaric acid); TS: total sugars (glucose plus fructose; g/L); ET: ethanol (% *v*/*v*); GLY: glycerol (g/L); AA: acetic acid (g/L). ^4^ nd: not determined.

**Table 2 foods-09-01549-t002:** Monitoring microbiological and oenochemical parameters during “Taurasi” (Aglianico of Taurasi (AGL)) experimental fermentations (data are mean ± standard deviation of two independent fermentations).

Fermentations (Starter Strain)	Time (Days)	Microbiological and Oenochemical Parameters						
		Yeast loads ^1^	pH	TA ^3^	TS ^3^	ET ^3^	GLY ^3^	AA ^3^
AGL1/AGL7 (R1)	0.5	7.2 ± 0.26 ^a 2^	3.35 ± 0.04 ^a^	4.91 ± 0.16 ^a^	209.00 ± 1.41 ^a^	nd ^4^	nd	nd
AGL2/AGL8 (R2)		6.57 ± 0.55 ^ab^	3.38 ± 0.02 ^a^	5.29 ± 0.05 ^a^	214.00 ± 2.83 ^a^	nd	nd	nd
AGL3/AGL9 (R3)		7.20 ± 0.34 ^a^	3.38 ± 0.00 ^a^	5.06 ± 0.16 ^a^	217.50 ± 2.12 ^a^	nd	nd	nd
AGL4/AGL10 (R4)		7.37 ± 0.15 ^a^	3.38 ± 0.00 ^a^	5.10 ± 0.10 ^a^	212.00 ± 2.83 ^a^	nd	nd	nd
AGL5/AGL6 (Control)		5.36 ± 0.69 ^b^	3.33 ± 0.01 ^a^	4.99 ± 0.16 ^a^	217.00 ± 4.24 ^a^	nd	nd	nd
AGL1/AGL7 (R1)	6.5	7.31 ± 0.04 ^a^	3.42 ± 0.04 ^a^	7.59 ± 0.18 ^abc^	18.29 ± 2.81 ^b^	11.42 ± 0.02 ^a^	7.29 ± 0.23 ^ab^	0.28 ± 0.02 ^a^
AGL2/AGL8 (R2)		6.93 ± 0.19 ^a^	3.44 ± 0.01 ^a^	7.03 ± 0.06 ^c^	26.14 ± 2.04 ^b^	11.23 ± 0.04 ^a^	6.96 ± 0.03 ^b^	0.21 ± 0.01 ^b^
AGL3/AGL9 (R3)		7.18 ± 0.23 ^a^	3.44 ± 0.02 ^a^	7.65 ± 0.13 ^ab^	23.41 ± 12.91 ^b^	11.28 ± 0.65 ^a^	7.65 ± 0.49 ^ab^	0.29 ± 0.00 ^a^
AGL4/AGL10 (R4)		7.10 ± 0.14 ^a^	3.45 ± 0.01 ^a^	7.72 ± 0.21 ^a^	18.78 ± 6.77 ^b^	11.40 ± 0.39 ^a^	8.60 ± 0.09 ^a^	0.20 ± 0.00 ^b^
AGL5/AGL6 (Control)		7.28 ± 0.14 ^a^	3.38 ± 0.01 ^a^	7.06 ± 0.13 ^bc^	113.20 ± 10.82 ^a^	5.90 ± 0.47 ^b^	4.82 ± 0.64 ^c^	0.14 ± 0.01 ^c^
AGL1/AGL7 (R1)	12.5	6.75 ± 0.34 ^a^	3.49 ± 0.03 ^a^	7.16 ± 0.17 ^abc^	2.03 ± 0.08 ^a^	11.82 ± 0.01 ^b^	8.83 ± 0.03 ^d^	0.30 ± 0.01 ^ab^
AGL2/AGL8 (R2)		6.28 ± 0.18 ^a^	3.40 ± 0.22 ^a^	6.73 ± 0.04 ^c^	1.82 ± 0.11 ^a^	12.46 ± 0.20 ^a^	9.16 ± 0.02 ^cd^	0.27 ± 0.01 ^bc^
AGL3/AGL9 (R3)		6.73 ± 0.20 ^a^	3.53 ± 0.01 ^a^	7.13 ± 0.09 ^bc^	1.74 ± 0.06 ^a^	12.29 ± 0.08 ^ab^	9.50 ± 0.13 ^bc^	0.35 ± 0.01 ^a^
AGL4/AGL10 (R4)		6.44 ± 0.06 ^a^	3.54 ± 0.05 ^a^	7.40 ± 0.27 ^ab^	1.81 ± 0.21 ^a^	12.20 ± 0.04 ^ab^	10.62 ± 0.21 ^a^	0.24 ± 0.02 ^c^
AGL5/AGL6 (Control)		7.02 ± 0.09 ^a^	3.47 ± 0.01 ^a^	7.74 ± 0.04 ^a^	2.25 ± 0.17 ^a^	12.06 ± 0.26 ^ab^	9.90 ± 0.11 ^b^	0.25 ± 0.01 ^c^

^1^ Log CFU/mL (determined on WL Nutrient Agar; Oxoid). ^2^ Data significance for columns and sampling time (ANOVA: Tukey *t*-test. *p* < 0.05. XLStat). ^3^ TA: total acidity (g/L of tartaric acid); TS: total sugars (glucose plus fructose; g/L); ET: ethanol (% *v*/*v*); GLY: glycerol (g/L); AA: acetic acid (g/L). ^4^ nd: not determined.

**Table 3 foods-09-01549-t003:** Number of yeast cultures isolated during the fermentation and identification by molecular methods.

Must	Fermentation (Starter Strain)	Days of Fermentation						Total N. of Isolates	N. of *S. cerevisiae* ^1^	N. of Non-*Saccharomyces* Strains (Species) ^3^	
		0	5		10		20				
Greco di Tufo	GR1 (W1)	7 (1) ^2^	4		5		5	21	20	1	(*M. pulcherrima*)
	GR7 (W1)	4	4		5		4	17	17		-
	GR2 (W2)	4 (1)	4		4		5	17	16	1	(*M. pulcherrima*)
	GR8 (W2)	3	5		5		3	16	16		-
	GR3 (W3)	5 (2)	4		5		6	20	18	2	(*P. occidentalis*; *H. uvarum*)
	GR9 (W3)	5 (1)	4		4		3	16	15	1	(*M. pulcherrima*)
	GR4 (W4)	6 (1)	4		5		4	19	18	1	(*P. anomala*)
	GR10 (W4)	3 (1)	5		5		5	18	17	1	(*P. fermentans*)
	GR5 (Contr.)	6 (6)	4		4		5	19	13	6	(*P. fermentans*; *H. uvarum*)
	GR6 (Contr.)	6 (6)	5		5		3	19	13	6	(*P. fermentans*)
								182	163	19	
**Must**	**Fermentation (Starter Strain)**	**Days of Fermentation**						**Total N. of Isolates**	**N. of *S. cerevisiae*** ^1^	**N. of Non-*Saccharomyces* Strains (Species)**	
		**0.5**		**6.5**		**12.5**					
Aglianico	AGL1 (R1)	5 (1)		3		3		11	10	1	(*St.la bacillaris*)
	AGL7 (R1)	3 (3)		3 (1)		3		9	5	4	( *P. fermentans, St.la bacillaris*)
	AGL2 (R2)	6 (3)		4		3		13	10	3	(*M. pulcherrima*; *St.la bacillaris*)
	AGL8 (R2)	4 (4)		3		5		12	8	4	( *H. uvarum*)
	AGL3 (R3)	4		4		3		11	11		-
	AGL9 (R3)	5 (4)		4		4		13	9	4	( *H. uvarum*)
	AGL4 (R4)	5 (1)		3		4		12	11	1	(*P. fermentans*)
	AGL10 (R4)	6 (1)		4		4		14	13	1	(*H. uvarum*)
	AGL5 (Contr.)	3 (2)		5 (1)		4		12	9	3	(*P. fermentans*; *H. uvarum*; *St.la bacillaris*)
	AGL6 (Contr.)	3 (3)		4		5		12	9	3	(*St.la bacillaris*; *H. uvarum*)
								119	95	24	

^1^ Identified by *Hae*III Internal Transcribed Spacer-Restriction Fragment Length Polymorphism. ^2^ Number of isolates at each phase (parenthesis the number of non-*Saccharomyces* strains). ^3^ Identified by ITS sequencing.

**Table 4 foods-09-01549-t004:** Number and types of Interdelta biotypes recorded in *S. cerevisiae* during fermentation of GR and AGL musts.

Must	Fermentation (Starter Strain)	Days of Fermentation					
		0	5		10		20
**GR**	GR1 (W1)	6 * (6a) **	4 (4a)		5(5a)		5 (3a, 1b, 1c)
	GR7 (W1)	4 (4a)	4 (3a, 1d)		5 (3a, 2e)		4 (4a)
	W1 % of dominance	100.0	87.5		80.0		77.8
	GR2 (W2)	4 (3a)	4 (3a, 1b)		4 (4a)		5 (4a, 1c)
	GR8 (W2)	3 (2a, 1d)	5 (5a)		5 (5a)		3 (2a, 1e)
	W2 % of dominance	83.3	88.9		100.0		75.0
	GR3 (W3)	3 (2a, 1b)	4 (3a, 1c)		5 (3a, 1d, 1e)		6 (4a, 1e; 1f)
	GR9 (W3)	4 (4a)	4 (2a, 1g, 1h)		4 (3a, 1i)		3 (3a)
	W1 % of dominance	85.7	62.5		66.7		77.8
	GR4 (W4)	5 (5a)	4 (2a, 2b)		5 (2a, 2c, 1d)		4 (3a, 1e)
	GR10 (W4)	2 (2a)	5 (4a, 1f)		6 (3a, 1f, 1g)		5 (3a, 1h, 1i)
	W4 % of dominance	100.0	66.7		45.5		66.7
	GR5 (Contr.)	-	4 (a–d)		4 (e–h)		5 (i–m)
	GR6 (Contr.)	-	5 (n–r)		5 (a, s–v)		3 (w–y)
**Must**	**Fermentation (Starter Strain)**	**Days of Fermentation**					
		**0.5**		**6.5**		**12.5**	
**AGL**	AGL1 (R1)	4 * (4a) **		3 (2a; 1b)		3 (2a; 1c)	
	AGL7 (R1)	-		2 (2a)		2 (2a)	
	R1 % of dominance	50.0		80.0		80.0	
	AGL2 (R2)	3 (3a)		4 (3a; 1b)		2 (2a)	
	AGL8 (R2)	-		3 (3a)		5 (3a; 1c; 1d)	
	R2 % of dominance	50.0		85.7		71.4	
	AGL3 (R3)	4 (4a)		4 (3a; 1b)		3 (3a)	
	AGL9 (R3)	1 (1a)		4 (4a)		4 (4a)	
	R3 % of dominance	100.0		87.5		100.0	
	AGL4 (R4)	4 (4a)		3 (1a; 2b)		4 (2a; 1b; 1c)	
	AGL10 (R4)	5 (4a, 1b)		4 (3a, 1d)		4 (2d; 2e)	
	R4 % of dominance	88.9		57.1		25.0	
	AGL5 (Contr.)	-		4 (2a; 1b; 1c)		4 (1d; 1e; 1f; 1g)	
	AGL6 (Contr.)	-		4 (1h; 1i; 1l; 1m)		5 (1h; 1i; 1n; 1o; 1p)	

* Number of isolates analyzed. ** Number of biotypes showing the same Interdelta patterns (biotypes coded with “a” exhibited the same Interdelta pattern of the strain used as starter). Biotype codes are related to each type of fermentation (yeast strain and control).

**Table 5 foods-09-01549-t005:** Preliminary characterization of wild and commercial (W1–W4) *S. cerevisiae* strains.

ID	^a^ Strain	^b^ Ethanol Tolerance (%)	^c^ H_2_S Production	^d^ Antagonistic Activity	^e^ Growth in SM at 14 °C	
					°Brix Reduction	Type of Growth
1	GR5-T5-61	15	2.5	+	3.85	S
2	GR5-T5-62	15	1.5	−	3.25	S
3	GR5-T5-63	14	2.0	−	2.45	D
4	GR6-T5-61	14	2.0	−	2.85	S
5	GR6-T5-63	14	2.0	−	2.55	S
6	GR6-T5-64	15	3.5	++	2.45	D
7	GR6-T5-65	15	2.5	+	2.55	S
8	GR5-T10-61	14	2.5	−	2.95	S
9	GR5-T10-63	14	2.5	−	4.05	S
10	GR5-T10-64	15	3.0	++	3.05	D
11	GR6-T10-51	14	1.0	−	3.35	S
12	GR6-T10-52	15	3.5	++	2.65	S
13	GR6-T10-53	14	2.0	−	2.55	S
14	GR6-T10-61	14	2.0	+	2.85	S
15	GR5-T20-51	14	2.5	−	2.40	S
16	GR5-T20-52	15	2.0	−	4.15	S
17	GR5-T20-53	14	2.5	−	2.35	S
18	GR6-T20-51	14	1.5	−	2.75	S
19	GR6-T20-53	14	2.5	±	2.85	S
20	GR5-T10-62	14	1.5	−	3.25	S
26	W1	14	2.5	+	3.35	S
27	GR1-T20-53	14	4.5	++	2.85	S
29	GR7-T5-64	13	4.0	++	4.35	D
30	GR7-T10-61	14	3.0	+	2.55	D
31	W2	13	2.5	+	2.35	S
32	GR2-T5-61	13	2.5	−	2.35	S
33	GR2-T20-55	13	2.5	−	3.15	S
34	GR8-T0-51	13	3.0	+	2.85	D
35	GR8-T20-42	14	3.0	+	2.75	D
36	W3	15	2.5	++	2.85	D
37	GR3-T0-54	14	2.5	−	2.45	S
38	GR3-T5-62	14	1.5	−	3.35	S
39	GR3-T10-61	15	1.5	++	2.95	S
40	GR3-T10-64	15	3.0	−	2.85	D
41	GR3-T20-55	15	5.0	−	3.55	D
42	GR9-T5-63	15	3.0	−	3.35	D
43	GR9-T5-64	15	1.5	++	3.15	S
44	GR9-T10-63	12	4.0	−	4.35	S
45	W4	15	5.0	−	3.35	S
46	GR4-T5-61	13	3.0	+	3.15	D
47	GR4-T10-63	15	4.5	−	2.85	S
48	GR4-T10-65	15	3.0	−	2.85	D
49	GR4-T20-53	15	3.5	+	2.75	S
50	GR10-T5-61	13	1.5	−	2.75	S
51	GR10-T10-65	14	3.5	++	3.35	S
52	GR10-T20-43	15	3.5	++	2.95	D
53	GR10-T20-44	13	2.0	−	2.95	D

^a^ Strain names are related to different fermentations (see Table 1 and Table 4). ^b^ In YPD (Yeast, Peptone, Dextrose) broth containing ethanol. ^c^ Estimated on Biggy agar (Oxoid). ^d^ Against Killer sensitive *S. cerevisiae* CECT1890. ^e^ In synthetic must (SM) at 14 °C: °Brix reduction, after 5 days; Type of growth: (D) dispersed, (S) growing mainly on the bottom of container.

**Table 6 foods-09-01549-t006:** Fermentation performances of selected wild *S. cerevisiae* strains versus commercial strains.

ID	Strains	FV ^1^	FP ^2^	Content of				pH	TA ^4^	Wine Color ^e^	
				AA ^3^	RS ^3^	GLY ^3^	ET ^3^			I	H
46	GR4-T5-61	0.95 ± 0.05 defg ^5^	7.63 ± 0.04 abc	0.15 ± 0.00 e	1.75 ± 1.07 ef	5.16 ± 0.41 bcdef	11.13 ± 1.03 a	3.08 ± 0.01 a	9.77 ± 0.02 c	1.03 ± 0.04 bcd	3.94 ± 0.07 ab
42	GR9-T5-63	1.47 ± 0.08 ab	7.60 ± 0.20 abc	0.36 ± 0.01 abcd	6.41 ± 2.19 bcde	4.81 ± 0.18 cdefgh	11.47 ± 1.11 a	3.09 ± 0.04 a	9.90 ± 0.21 bc	1.07 ± 0.07 bcd	3.83 ± 0.23 ab
6	GR6-T5-64	1.16 ± 0.05 bcdef	7.44 ± 0.02 abc	0.22 ± 0.04 de	6.47 ± 1.20 bcde	4.65 ± 0.30 efgh	11.12 ± 0.99 a	3.05 ± 0.05 a	11.53 ± 0.66 ab	1.19 ± 0.06 abc	3.49 ± 0.21 bcd
29	GR7-T5-64	1.29 ± 0.06 abcd	7.65 ± 0.06 abc	0.36 ± 0.01 abcd	3.14 ± 0.62 def	4.56 ± 0.19 efgh	11.72 ± 0.65 a	3.07 ± 0.01 a	9.85 ± 0.07 c	1.31 ± 0.15 ab	3.07 ± 0.07 cd
38	GR3-T5-62	1.20 ± 0.11 bcdef	7.63 ± 0.05 abc	0.49 ± 0.04 a	2.34 ± 1.19 ef	5.36 ± 0.20 bcd	11.44 ± 0.23 a	3.13 ± 0.00 a	9.43 ± 0.18 c	1.06 ± 0.02 bcd	3.86 ± 0.12 ab
2	GR5-T5-62	0.88 ± 0.01 fg	7.61 ± 0.03 abc	0.34 ± 0.00 abcd	2.60 ± 1.55 ef	6.60 ± 0.05 a	11.43 ± 0.22 a	3.08 ± 0.01 a	10.74 ± 0.18 abc	1.12 ± 0.05 abcd	3.75 ± 0.12 abc
7	GR6-T5-65	1.35 ± 0.10 abc	7.46 ± 0.04 abc	0.33 ± 0.00 bcd	8.33 ± 3.15 abcd	4.46 ± 0.13 gh	11.99 ± 0.09 a	3.06 ± 0.01 a	10.21 ± 0.12 bc	1.08 ± 0.05 bcd	3.85 ± 0.12 ab
1	GR5-T5-61	1.24 ± 0.08 bcde	7.44 ± 0.04 abc	0.36 ± 0.02 abcd	9.44 ± 3.04 abc	4.78 ± 0.14 defgh	11.44 ± 0.42 a	3.04 ± 0.04 a	10.76 ± 0.06 abc	1.03 ± 0.00 bcd	4.01 ± 0.07 ab
43	GR9-T5-64	0.96 ± 0.03 defg	7.65 ± 0.04 abc	0.33 ± 0.01 bcd	4.57 ± 1.32 cdef	4.37 ± 0.06 h	11.67 ± 0.69 a	3.09 ± 0.01 a	10.16 ± 0.11 bc	1.09 ± 0.01 bcd	3.89 ± 0.08 ab
10	GR5-T10-64	1.39 ± 0.07 ab	7.65 ± 0.22 abc	0.22 ± 0.10 de	0.50 ± 0.00 f	4.50 ± 0.05 fgh	11.17 ± 2.08 a	3.05 ± 0.01 a	9.65 ± 0.07 c	1.14 ± 0.06 abcd	3.61 ± 0.17 bc
44	GR9-T10-63	0.77 ± 0.13 g	7.80 ± 0.04 ab	0.42 ± 0.00 ab	2.94 ± 1.34 def	5.49 ± 0.10 ab	11.82 ± 0.58 a	3.07 ± 0.04 a	10.11 ± 0.35 bc	1.22 ± 0.15 abc	2.80 ± 0.23 d
9	GR5-T10-63	1.60 ± 0.00 a	7.71 ± 0.13 ab	0.36 ± 0.05 abcd	0.50 ± 0.00 f	5.24 ± 0.18 bcde	12.14 ± 1.08 a	3.07 ± 0.03 a	10.09 ± 0.05 bc	0.87 ± 0.02 d	4.36 ± 0.06 a
51	GR10-T10-65	0.95 ± 0.02 defg	7.38 ± 0.04 abc	0.36 ± 0.01 abcd	10.75 ± 0.64 ab	4.94 ± 0.03 bcdefgh	11.58 ± 0.58 a	3.04 ± 0.03 a	10.84 ± 0.47 abc	1.04 ± 0.06 bcd	3.95 ± 0.2 ab
41	GR3-T20-55	1.41 ± 0.05 ab	7.76 ± 0.08 ab	0.39 ± 0.01 ab	0.50 ± 0.00 f	6.26 ± 0.19 a	11.56 ± 0.56 a	3.11 ± 0.01 a	10.14 ± 0.03 bc	1.03 ± 0.09 bcd	3.54 ± 0.48 bc
52	GR10-T20-43	1.01 ± 0.13 cdefg	7.65 ± 0.07 abc	0.22 ± 0.11 cde	0.50 ± 0.00 f	4.69 ± 0.00 defgh	11.65 ± 0.63 a	3.05 ± 0.01 a	9.69 ± 0.03 c	1.04 ± 0.01 bcd	3.71 ± 0.26 abc
16	GR5-T20-52	1.42 ± 0.00 ab	7.80 ± 0.12 ab	0.37 ± 0.05 abcd	0.50 ± 0.00 f	5.52 ± 0.01 b	11.20 ± 0.50 a	3.05 ± 0.01 a	10.87 ± 0.00 abc	1.07 ± 0.03 bcd	3.84 ± 0.12 ab
49	GR4-T20-53	1.31 ± 0.01 abc	7.08 ± 0.04 c	0.38 ± 0.01 abc	12.78 ± 0.35 a	4.83 ± 0.06 bcdefgh	11.23 ± 0.58 a	3.14 ± 0.13 a	12.03 ± 1.52 a	1.05 ± 0.08 bcd	3.76 ± 0.11 abc
26	W1	0.80 ± 0.06 g	7.21 ± 0.08 bc	0.34 ± 0.02 abcd	10.85 ± 0.55 ab	5.10 ± 0.21 bcdefg	11.24 ± 0.31 a	3.07 ± 0.03 a	10.37 ± 0.29 abc	1.04 ± 0.04 bcd	3.89 ± 0.10 ab
31	W2	0.93 ± 0.08 efg	7.67 ± 0.03 ab	0.15 ± 0.00 e	0.50 ± 0.00 f	6.87 ± 0.24 a	11.55 ± 0.70 a	3.07 ± 0.01 a	9.79 ± 0.11 c	1.11 ± 0.15 bcd	3.82 ± 0.15 ab
36	W3	1.05 ± 0.16 cdefg	7.84 ± 0.20 a	0.32 ± 0.00 bcd	1.72 ± 1.02 ef	4.52 ± 0.08 fgh	11.43 ± 1.25 a	3.06 ± 0.01 a	10.12 ± 0.00 bc	1.40 ± 0.06 a	3.91 ± 0.03 ab
45	W4	0.95 ± 0.15 defg	7.60 ± 0.22 abc	0.31 ± 0.03 bcde	3.67 ± 1.79 def	5.02 ± 0.15 bcdefgh	11.88 ± 0.76 a	3.05 ± 0.00 a	10.22 ± 0.45 bc	1.01 ± 0.02 cd	3.96 ± 0.10 ab
Average values	Total	1.15 ± 0.24	7.58 ± 0.19	0.32 ± 0.09	4.32 ± 3.99	5.13 ± 0.70	11.52 ± 0.29	3.07 ± 0.03	10.30 ± 0.64	1.10 ± 0.11	3.75 ± 0.33
	Wild	1.20 ± 0.24	7.58 ± 0.18	0.33 ± 0.08	4.35 ± 3.98	5.07 ± 0.63	11.52 ± 0.30	3.07 ± 0.03	10.34 ± 0.70	1.08 ± 0.10	3.72 ± 0.36
	Commercial	0.93 ± 0.10	7.58 ± 0.27	0.28 ± 0.09	4.19 ± 4.63	5.38 ± 1.03	11.53 ± 0.27	3.06 ± 0.01	10,13 ± 0.25	1.14 ± 0.18	3.90 ± 0.06

^1^ FV, Fermentation vigor: grams of CO_2_ produced in 80 mL of must in the first 3 days of fermentation. ^2^ FP, Fermentation power: total grams of CO_2_ produced in 80 mL at the end of fermentation. ^3^ AA, acetic acid (g/L); RS, residual sugars (g/L); GLY, glycerol (g/L), ET, ethanol (% vol/vol); determined by HPLC (High Performance Liquid Chromatography) at the end of fermentation. ^4^ TA, Total acidity:g/L of tartaric acid (25 mL of wine and 0.25 N NaOH) reached at the end of fermentation. ^e^ Spectrophotometric determinations: I, intensity (Absorbance A420 + A520 + A620); H, hue (A420/A520). ^5^ Data significance for columns (ANOVA: Tukey *t*-test. *p* < 0.05. XLStat).

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
