# Peer review of "Dominance of S. cerevisiae Commercial Starter Strains during Greco di Tufo and Aglianico Wine Fermentations and Evaluation of Oenological Performances of Some Indigenous/Residential Strains"

_foods, 2020, doi:10.3390/foods9111549_

Round 1

Reviewer 1 Report

Manuscript Number: foods-954965

Title: Dominance of S. cerevisiae commercial starter strains during Greco di Tufo and Aglianico wine fermentations and comparison with oenological performances exhibited by indigenous/residential strains

Author(s): Aponte et al.

The paper “Dominance of S. cerevisiae commercial starter strains during Greco di Tufo and Aglianico wine fermentations and comparison with oenological performances exhibited by indigenous/residential strains” reports results about the dominance S. cerevisiae commercial starter during fermentation of two grapevine varieties. In addition, these yeasts are compared with indigenous strains. The study is interesting; however, the results and/or the trials are not always clearly explained; therefore, certain aspects should be clarified. Specifically, more colonies should be analysed to elucidate the implantation ability of the starters and the genetic identity of isolates should be indicated. Moreover, the criteria used for strain selection are not included. Therefore, I recommend the acceptation of this paper after major revision.

 Comments and suggestions:

Title:

The title is too long and may confuse the reader: the dominance of indigenous strains was not evaluated. I mean, they have not been added as starters, have they?

Abstract:

Line 19-20: During the fermentations of Greco di Tufo (GR) and Aglianico of Taurasi (AGL) what? Grapes, must?

Paragrapgh 24-28: A high genetic diversity within S. cerevisiae strains was detected...I would include different sentences for starter-led fermentations and spontaneous fermentations. Delete b) native

26 strains isolated from replicates of the same fermentation showed different genetic profiles; this indicate that they are not good replicates…in this case, I think the number of colonies isolated and screened was not enough (is the main weak point of this study)

Lines 31-32: The study further highlighted the low dominance of some commercial starter cultures. Moreover, autochthonous yeast strains proved to be sometimes more aggressive in terms of implantation-As mentioned previously is not clear in the manuscript that indigenous strains were used as starters, and the isolates number should be higher to confirm these sentences.

Keywords. Dominace /implanation- revise

Introduction:

Line 82: S. cerevisiae strains (from “Greco di Tufo”). What about strains from Aglianico of Taurasi?

Materials and methods:

Line 112. Did you follow the fermentation kinetics? How? Do you have the initial characteristics of musts

Lines 136-137. How many colonies were isolated per sample? If your interest was S. cerevisiae, you should have isolated a representative number of colonies of them, in addition to those of different morphology.

Lines 146-147. I wonder why did you use strains from GR fermentations and not from AGL fermentations. Have GR strains with similar genetic profiles to AGL strains?

Line 194. differences of fermentations with different strains? Please clarify

 Results:

Figure 1 to 3. I am surprised with the strain order

Lines 209-210. mtDNA-RFLPs for S. cerevisiae strains differentiation works better with Hinf I.

May be you could include only fig 1, because is the method you used later for differentiation, and include fig 2 a fig 3 as supplementary material

Line 257. Starmerella (St.la) bacillaris. Please, revise the abbreviation of this species St? Check Table 3.

Line 259. AGR7 and AGR5...Do you mean AGL7 and AGL5?

3.4. Strains tracking by molecular typing. Definitely a higher number of isolates from each fermentation and phase should have been checked to draw conclusions

Table 4. Please revise nomenclature to clarify this table. In the foot table you indicate that biotypes coded with “a” exhibited the same Interdelta pattern of the strain used as starter, for example for Gr1 and GR7. What about GR2 and GR8? Here a is different from the previous one, isn`t it? Idem for the remaining letters. How can we differentiate all the strains? Were the indigenous strains in GR1 the same as those found in GR3 or in control fermentations?. Idem for AGL fermentations.

How many different strains, or interdelta patterns, did you find in total? Please clarify this point

 3.5 Technological characterization of yeast cultures from Greco di Tufo. Why did you decide to use these cultures? Were all of them different strains from a genetic point of view? Were they similar to those strains found in AGL fermentations? Did you find common strains?

Lines 331-332. Which criteria did you follow to choose these 21 strains? which was the frequency of these strains in 100L trials?

 Discussion:

Lines 413-414: You should mention in results global data about diversity in each fermentation and in total, this is why it is important to know if strains from different fermentations show the same genetic patterns.

Conclusions:

Line 433- The dominance

Paragraph 434-439: it is fine, but unexpected. I think you should indicate clear the aim of this study at the end of the introduction. Idem in the title. You start evaluating the implantation of starters as the main objective, and selection of indigenous strains is secondary; however, by the end the second part becomes the protagonist.

References:

Line 516: didigest?

Line 517. Ref 26. .in therms

Revise the references, sometimes the name of species is not in italics

Author Response

Dear Reviewer 1

Please find below a report, point by point, as the manuscript was modified following your comments/suggestion

Many thanks for spending your time for revising our work

Comments and suggestions:

Title: 

Q1_Rev.: The title is too long and may confuse the reader: the dominance of indigenous strains was not evaluated. I mean, they have not been added as starters, have they?

Auth.: Thank you for suggestion. The title was modified

Abstract:

Q2_Rev.: Line 19-20: During the fermentations of Greco di Tufo (GR) and Aglianico of Taurasi (AGL) what? Grapes, must?

Auth.: specified

Q3_Rev.: Paragrapgh 24-28: A high genetic diversity within S. cerevisiae strains was detected...I would include different sentences for starter-led fermentations and spontaneous fermentations. Delete b) native

Auth.: sentences were modified according reviewer suggestion

Q4_Rev.: 26 strains isolated from replicates of the same fermentation showed different genetic profiles; this indicate that they are not good replicates…in this case, I think the number of colonies isolated and screened was not enough (is the main weak point of this study)

Auth.: the procedure of isolation was specified in M&M section (paragraph 2.3). We think that if the sampling of the colonies was not done well it must be the opposite (less diversity). The high diversity found is a consequence of the accurate method of colonial isolation and the remarkable genetic diversity of S. cerevisiae.

Q5_Rev.: Lines 31-32: The study further highlighted the low dominance of some commercial starter cultures. Moreover, autochthonous yeast strains proved to be sometimes more aggressive in terms of implantation-As mentioned previously is not clear in the manuscript that indigenous strains were used as starters, and the isolates number should be higher to confirm these sentences.

Auth.: “terms of implantation” was replaced with “terms of fermentation vigour in GR must

Keywords. 

Q6_Rev.: Dominace /implanation- revise

Auth.: revised

Introduction:

Q7_Rev.: Line 82: S. cerevisiae strains (from “Greco di Tufo”). What about strains from Aglianico of Taurasi?

Auth.: As specified in this sentence and now also in the title, we have only dealt with a group of strains. There is no reason. In other research we have worked with indigenous strains from Aglianico.

Materials and methods:

Q8_Rev.: Line 112. Did you follow the fermentation kinetics? How? Do you have the initial characteristics of musts

Auth.: Microbiological and oenochemical parameters were monitored during fermentations and data are reported in table 1

Q9_Rev.: Lines 136-137. How many colonies were isolated per sample? If your interest was S. cerevisiae, you should have isolated a representative number of colonies of them, in addition to those of different morphology.

Auth.: The procedure of isolation was specified in M&M section (paragraph 3.3). “Musts and wines were serially diluted in quarter strength Ringer’s solution (Oxoid) and spread-plated on WL-nutrient agar (Oxoid). After incubation at 28°C for 5 days, plates were used for viable counts and yeasts isolation. WL is a non-selective medium that allows to discriminate wine yeasts species and strains and can be profitably used for monitoring the yeast population dynamics during wine fermentation (Pallmann et al., 2001). In order to analyse the dominant cultivable yeast microbiota, colonies showing different morphology and color on plates with 15-150 colonies were isolated and purified on WL nutrient agar. In the isolation phase an attempt was made to maintain the proportions of the different types of colonies.”

Q10_Rev.: Lines 146-147. I wonder why did you use strains from GR fermentations and not from AGL fermentations.

Auth.: please see answer Q7

Q11_Rev.: Have GR strains with similar genetic profiles to AGL strains?

Auth.: this comparison was not made. since we had decided to evaluate only the preformance of isolates from GR

Q12_Rev.: Line 194. differences of fermentations with different strains? Please clarify

Auth.: the sentence was modified (paragraph 2.7)

Results:

Q13_Rev.: Figure 1 to 3. I am surprised with the strain order

Auth.: Why?

Q14_Rev.: Lines 209-210. mtDNA-RFLPs for S. cerevisiae strains differentiation works better with Hinf I.

Auth.: I agree. We had thought of using other enzymes such as Hinf I. However as the other markers had given good results we did not. Furthermore, this marker was not used to characterize isolates during fermentations

Q15_Rev.: May be you could include only fig 1, because is the method you used later for differentiation, and include fig 2 a fig 3 as supplementary material

Auth.: We agree with your suggestion. Figures 2 and 3 will be presented as supplementary material

Q16_Rev.: Line 257. Starmerella (St.labacillaris. Please, revise the abbreviation of this species St? Check Table 3.

Auth.: revised

Q17_Rev.: Line 259. AGR7 and AGR5...Do you mean AGL7 and AGL5?

Auth.: revised

Q18_Rev.: 3.4. Strains tracking by molecular typing. Definitely a higher number of isolates from each fermentation and phase should have been checked to draw conclusions

Auth.: see answers to questions Q4 and Q9

Q19_Rev.: Table 4. Please revise nomenclature to clarify this table. In the foot table you indicate that biotypes coded with “a” exhibited the same Interdelta pattern of the strain used as starter, for example for Gr1 and GR7. What about GR2 and GR8? Here a is different from the previous one, isn`t it? Idem for the remaining letters. How can we differentiate all the strains? Were the indigenous strains in GR1 the same as those found in GR3 or in control fermentations?. Idem for AGL fermentations.

How many different strains, or interdelta patterns, did you find in total? Please clarify this point

Auth.: it was specified in the foot note of the table that “Biotype codes are related to each type of fermentation (yeast strain and control). Genetic profiles of strains coming from different type of fermentations were (yeast strain and control) and different cultivars were not compared.

Q20_Rev.: 3.5 Technological characterization of yeast cultures from Greco di Tufo. Why did you decide to use these cultures?

Auth.: please see answer Q7

Q21_Rev.: Were all of them different strains from a genetic point of view? Were they similar to those strains found in AGL fermentations? Did you find common strains?

Auth.: Genetic profiles of strains coming from different type of fermentations were (yeast strain and control) and different cultivars were not compared.

Q22_Rev.: Lines 331-332. Which criteria did you follow to choose these 21 strains?

Auth.: on the basis of preliminary characterization [table 5 and fig 3 (ex fig 5)]

Q23_Rev.: which was the frequency of these strains in 100L trials?

Auth.: on the basis of code names reported in table 3 we can obtain these data. But what good would it do?

Discussion:

Q24_Rev.: Lines 413-414: You should mention in results global data about diversity in each fermentation and in total, this is why it is important to know if strains from different fermentations show the same genetic patterns.

Auth.: The sentence was modified

Conclusions: 

Q25_Rev.: Line 433- The dominance

Auth.: corrected

Q26_Rev.: Paragraph 434-439: it is fine, but unexpected. I think you should indicate clear the aim of this study at the end of the introduction. Idem in the title. You start evaluating the implantation of starters as the main objective, and selection of indigenous strains is secondary; however, by the end the second part becomes the protagonist.

Auth.: title and end of introduction was modified

References:

Q27_Rev.: Line 516: didigest?

Auth.: corrected

Q28_Rev.: Line 517. Ref 26. .in therms

Auth.: corrected

Q29_Rev.: Revise the references, sometimes the name of species is not in italics

Auth.: revised

Reviewer 2 Report

The manuscript called “Dominance of S. cerevisiae commercial starter strains during Greco di Tufo and Aglianico wine fermentations and comparison with oenological performances exhibited by indigenous/residential strains” by  Aponte et al. is an interesting ecological study on the competition between commercial and indigenous wine yeasts. The approach is not new, and some of the conclusions have been found in the past, but the authors have extensively characterized new Greco di Tufo strains that are promising targets for “pied de cuve” production. It is a pity that Aglianico strains were not fully characterized, particularly the strains in AGL10-12.5 that out compete the commercial yeast R4. Commercial names of W1-W4 and R1-R4 should be fully disclosed to provide a better understanding of the characteristics of the local strains by comparing both. I understand that W are fit to ferment white wine, and R, red wine. Authors find that indigenous yeast have higher FV in average, and I agree that may reflect better adaptation. However, there is no any indication that strains isolated form inoculated fermentations behave better on average. Authors indicate that GR5-T10-63 is a good candidate to be produced as PdC, but this is a strain isolated from spontaneous fermentation. It would be interesting to include in Table 6 average levels for commercial, GR5+GR6 strains and the rest. The Discussion has too much information from other authors findings and lacks more comments on the actual data. Not much attention is paid to the non-conventional yeasts. Comment on S. bacillaris at 6.5 days of fermentation.

Minor points.

Line 213. I would combine Figures 1 and 2 in one and send Figure 3 (which quality is not good) to Supplementary Materials.

Line 240. “182 and 119 yeast cultures…..”

Table 1. Why these GR fermentations do not achieve dryness at 21 days and many of Table 6 did? Indicate the time fermentation was ended.

Figure 5. What does the strains in bold mean?

Author Response

Dear Reviewer 2

Please find below a report, point by point, as the manuscript was modified following your comments/suggestion

Many thanks for spending your time for revising our work

Reviewer 2

Comments and Suggestions for Authors

The manuscript called “Dominance of S. cerevisiae commercial starter strains during Greco di Tufo and Aglianico wine fermentations and comparison with oenological performances exhibited by indigenous/residential strains” by  Aponte et al. is an interesting ecological study on the competition between commercial and indigenous wine yeasts. The approach is not new, and some of the conclusions have been found in the past, but the authors have extensively characterized new Greco di Tufo strains that are promising targets for “pied de cuve” production.

Q1_Rev.: It is a pity that Aglianico strains were not fully characterized, particularly the strains in AGL10-12.5 that out compete the commercial yeast R4.

Auth.: As specified in the last sentence of introduction and now also in the title, we have only dealt with a group of strains. There is no reason. In other research we have worked with indigenous strains from Aglianico.

Q2_Rev.: Commercial names of W1-W4 and R1-R4 should be fully disclosed to provide a better understanding of the characteristics of the local strains by comparing both.

Auth.: We did not consider it appropriate to publish the name of the strains. Also because some of them did not show high dominance. It calls to favor someone and penalize someone else

Q3_Rev.: I understand that W are fit to ferment white wine, and R, red wine.

Auth.: Yes. Specified in section 2.1.

Q4_Rev.: Authors find that indigenous yeast have higher FV in average, and I agree that may reflect better adaptation. However, there is no any indication that strains isolated form inoculated fermentations behave better on average. Authors indicate that GR5-T10-63 is a good candidate to be produced as PdC, but this is a strain isolated from spontaneous fermentation. It would be interesting to include in Table 6 average levels for commercial, GR5+GR6 strains and the rest.

Auth.: in table 6 the average values of the natural and commercial strains have been added (two new lines).

Now comparing the FV data of commercial and wild strains it is clear that most of the wild strains showed a higher FV value than commercial strains. Some of these strains were isolated from vinifications inoculated with commercial strains (GR7; GR3/GR9 and GR4/GR10)

Q5_Rev.: The Discussion has too much information from other authors findings and lacks more comments on the actual data. Not much attention is paid to the non-conventional yeasts. Comment on S. bacillaris at 6.5 days of fermentation.

Auth.: in the discussion section some sentences were added regarding the occurrence and the importance of S. bacillaris 

Minor points.

Q6_Rev.: Line 213. I would combine Figures 1 and 2 in one and send Figure 3 (which quality is not good) to Supplementary Materials.

Auth.: as also reviewer 1 suggested figures 2 and 3 were moved in the Supplementary Materials

Q7_Rev.: Line 240. “182 and 119 yeast cultures…..”

Auth.: modifies

Q8_Rev.: Table 1. Why these GR fermentations do not achieve dryness at 21 days and many of Table 6 did? Indicate the time fermentation was ended.

Auth.: we think that data are not comparable. Table 1 refer to wine fermentations with a native microflora, Table 6 refer to microvinifications in “steril must” conditions.

Q9_Rev.: Figure 5. What does the strains in bold mean?

Auth.: specified in the foot note.

Reviewer 3 Report

General comments:

The study by Aponte and collaborators deals with the dominance of commercial starter cultures in competition with indigenous yeasts during the fermentation of white and red grape must. The research approach is not quite new but experimentally well planned using appropriate analytical methods and modern molecular techniques to identify the yeasts. The results obtained seem to be quite relevant for wine production.

However, there are weaknesses in the presentation. Most of the tables appear as a representation of raw data and blur the essence of the study. Especially table 4 is extremely difficult to read, probably only for the authors. I suggest adding the superfluous data series as a supplement and streamlining the manuscript.

Furthermore, there are some inaccuracies in the species designation. In line 200, 243 and 262 it is stated that the amplicon size of 850 bp identified the isolates as Saccharomyces spp. However, this also includes the species S. bayanus. Recently it has also been repeated shown that in spontaneous fermentations often hybrids of S. cerevisiae, S. bayanus and S. kudriavzevii dominate (Fermentation 4, 67; doi: 10.3390/fermentation4030067). How can we be sure that in figure 1 and following we are dealing with different strains of S. cerevisiae or different S. species? The point should be clarified. Otherwise the text should only refer to Saccharomyces spp. biotypes and not to S. cervisiae.

Minor Comments

Line

98 and afterwards: use g number instead of rpm if possible

101: supernatant

122: di-ammonium

126: rehydrated

159: please give a short explanation of the activity test, what is the principle of detection? If pH variation, why?

286: UFC or CFU?

291: could not found footnote 3 in the table

388: ADY?

Author Response

Dear Reviewer 3

Please find below a report, point by point, as the manuscript was modified following your comments/suggestion

Many thanks for spending your time for revising our work

General comments:

The study by Aponte and collaborators deals with the dominance of commercial starter cultures in competition with indigenous yeasts during the fermentation of white and red grape must. The research approach is not quite new but experimentally well planned using appropriate analytical methods and modern molecular techniques to identify the yeasts. The results obtained seem to be quite relevant for wine production.

Q1_Rev.: However, there are weaknesses in the presentation. Most of the tables appear as a representation of raw data and blur the essence of the study. Especially table 4 is extremely difficult to read, probably only for the authors. I suggest adding the superfluous data series as a supplement and streamlining the manuscript.

Auth.: We understand the difficulty of reading table 4. However, we have added an explanatory note. We hope it helps the reader. “Biotype codes are related to each type of fermentation (yeast strain and control)”. Genetic profiles of strains coming from different type of fermentations were (yeast strain and control) and different cultivars were not compared.

Q2_Rev.: Furthermore, there are some inaccuracies in the species designation. In line 200, 243 and 262 it is stated that the amplicon size of 850 bp identified the isolates as Saccharomyces spp. However, this also includes the species S. bayanus. Recently it has also been repeated shown that in spontaneous fermentations often hybrids of S. cerevisiae, S. bayanus and S. kudriavzevii dominate (Fermentation 4, 67; doi: 10.3390/fermentation4030067). How can we be sure that in figure 1 and following we are dealing with different strains of S. cerevisiae or different S. species? The point should be clarified. Otherwise the text should only refer to Saccharomyces spp. biotypes and not to S. cervisiae.

Auth.: Sorry but a sentence is missing in paragraph 3.1. It has been now added.

In fact, as reported in the M&M section, both commercial strains before their use (paragraphs 2.1) and all strains isolated during fermentation (paragraphs 2.3) were analyzed by ITS-RFLP. In addition, in table 3 it was specified in the foot not that S. cerevisiae strains were Identified by Hae III ITS-RFLP. Furthermore, the molecular markers Interdelta and DAN 4 are quite specific for S. cerevisiae. Therefore, we exclude the presence of other species

The mistake was also corrected in paragraph 3.3.

Round 2

Reviewer 1 Report

The revision of the paper “Dominance of S. cerevisiae commercial starter strains during Greco di Tufo and Aglianico wine fermentations and evaluation of oenological performances of some indigenous/residential strains isolated from Greco di Tufo” have been improved compared to the first manuscript. However, there is still a few points that could be modified.

 Comments and suggestions:

Title:

The new title is even longer… You could stop in strains: “Dominance of S. cerevisiae commercial starter strains during Greco di Tufo and Aglianico wine fermentations and evaluation of oenological performance of some indigenous/residential strains”

Keywords. implantaion- revise

Materials and methods:

Auth.: the procedure of isolation was specified in M&M section (paragraph 2.3). We think that if the sampling of the colonies was not done well it must be the opposite (less diversity). The high diversity found is a consequence of the accurate method of colonial isolation and the remarkable genetic diversity of S. cerevisiae.

Auth.:. In the isolation phase an attempt was made to maintain the proportions of the different types of colonies.”

As you mention WL media allow the differentiation of different yeast species based on their colony morphology, bit it is not useful to distinguished different S cerevisiae strains. So, if you are studying the diversity of S. cerevisiae strains you should isolate a representative number of colonies (with the characteristic morphology of this species) and them you should check the strain diversity.

 Results:

Figure 1 to 3. I am surprised with the strain order (3-2-4-1?) I usually follow the order 1-2-3-4-….

Line 210. S. cerevisiae (in italics). Revise this sentence…who?

3.4. Strains tracking by molecular typing.

As mention above a higher number of isolates from each fermentation and phase should have been checked to draw conclusions

It would have been very interesting to compare strains among fermentations and even between grape varieties

Q23_Rev.: which was the frequency of these strains in 100L trials?

Auth.: on the basis of code names reported in table 3 we can obtain these data. But what good would it do?

You are evaluating the performance of these strains at lab scale, but if you have the data about their presence/dominance at pilot scale, that is a good data about their real oenological performance

 References:

Revise the references, sometimes the name of species is not in italics (L 549,

Author Response

Dear Reviewer, 1

Please find below a report, point by point, as the manuscript was further modified following your comments/suggestion

Many thanks for spending your time for re-evaluating our work

Comments and Suggestions for Authors

The revision of the paper “Dominance of S. cerevisiae commercial starter strains during Greco di Tufo and Aglianico wine fermentations and evaluation of oenological performances of some indigenous/residential strains isolated from Greco di Tufo” have been improved compared to the first manuscript. However, there is still a few points that could be modified.

Comments and suggestions:

Q1_Rev.: Title: The new title is even longer… You could stop in strains: “Dominance of S. cerevisiae commercial starter strains during Greco di Tufo and Aglianico wine fermentations and evaluation of oenological performance of some indigenous/residential strains”

Auth.: modified as suggested

Q2_Rev.: Keywords. implantaion-revise

Auth.: revised

Materials and methods:

Auth.: the procedure of isolation was specified in M&M section (paragraph 2.3). We think that if the sampling of the colonies was not done well it must be the opposite (less diversity). The high diversity found is a consequence of the accurate method of colonial isolation and the remarkable genetic diversity of S. cerevisiae.

Auth.:. In the isolation phase an attempt was made to maintain the proportions of the different types of colonies.”

Q3_Rev.: As you mention WL media allow the differentiation of different yeast species based on their colony morphology, bit it is not useful to distinguished different S cerevisiae strains. So, if you are studying the diversity of S. cerevisiae strains you should isolate a representative number of colonies (with the characteristic morphology of this species) and them you should check the strain diversity.

Auth.: Excuse me for insisting, but on the WL substrate the colonies of S. cerevisiae are not all the same. In fact, the color of the colonies varies from cream to green (Pallmann et al., 2001- Am. J. Enol. Vitic. 52: 198-203). The size of the colonies is variable. The surface of the colonies is generally smooth and opaque, the consistency is generally creamy, but in flocculent strains it is consistent.

Results:

Q4_Rev.: Figure 1 to 3. I am surprised with the strain order (3-2-4-1?) I usually follow the order 1-2-3-4-….

Auth.: Unfortunately, the photo happened like this. we thought that putting the name of the strain could help the reader.

Of course, they can be modified. The options are 3:

1) replace the name of the strains with progressive numbers (1 .....) and explain in the legend what they are;

2) cut each line (profiles) and arrange them in sequence (R1, R2, R3, ..). Which I would never do because it would seem an artifact.

3) leave the figures as they are.

Tell me what I can do.

Q5_Rev.: Line 210. S. cerevisiae (in italics). Revise this sentence…who?

Auth.: revised

Q6_Rev.: 3.4. Strains tracking by molecular typing. As mention above a higher number of isolates from each fermentation and phase should have been checked to draw conclusions. It would have been very interesting to compare strains among fermentations and even between grape varieties

Auth.: strains among fermentations of Greco were compare by their technological traits and by their fermentation performances.

Of course, it would have been interesting to compare all strains of S. cerevisae isolated both genetically and physiologically, but we did not have the strength, at this time, to do so.

However, the purpose of the work was not the analysis of biodiversity, but to understand why the indigenous / residential strains, which at the beginning of fermentation are present in low concentrations, are able to compete and record similar levels of loads of the inoculated strains.

Q7_Rev.: (exQ23_Rev.): which was the frequency of these strains in 100L trials?

Auth.: on the basis of code names reported in table 3 we can obtain these data. But what good would it do?

You are evaluating the performance of these strains at lab scale, but if you have the data about their presence/dominance at pilot scale, that is a good data about their real oenological performance

Auth.: the indigenous/residential strains were isolated from counting plates (countable plates: 15-150 colonies), therefore were part of dominant yeast microflora in each phase. In the starter inoculated fermentations, they were co-dominant with commercial strain. However, I modified the first column of the tables 1 and 2, were yeast loads are reported. Moreover, I specified (foot note) in table 5 that Strain names are related at each fermentation (and isolation time) monitored.

References:

Q8_Rev.: Revise the references, sometimes the name of species is not in italics (L 549,

Auth.: revised

Reviewer 3 Report

The manuscript can be published in the present revised version.

Author Response

Dear Reviewer 3

Many thanks for spending your time for re-evaluating our work